# Does Back Squat Exercise Lead to Regional Hypertrophy among Quadriceps Femoris Muscles?

**DOI:** 10.3390/ijerph192316226

**Published:** 2022-12-04

**Authors:** Filip Kojic, Igor Ranisavljev, Milos Obradovic, Danimir Mandic, Vladan Pelemis, Milos Paloc, Sasa Duric

**Affiliations:** 1Department for Physical Education, Teacher Education Faculty, University of Belgrade, 11000 Belgrade, Serbia; 2Department for Strength and Conditioning Training, Faculty of Sport and Physical Education, University of Belgrade, 11000 Belgrade, Serbia; 3Sports Center, Department for University Sport, University of Belgrade, 11000 Belgrade, Serbia; 4Liberal Arts Department, American University of the Middle East, Egaila 54200, Kuwait

**Keywords:** knee extensors, resistance training, strength, cross-sectional area

## Abstract

The present study investigated effects of squat resistance training on intermuscular hypertrophy of quadriceps femoris muscles (i.e., rectus femoris, RF; vastus intermedius, VI; vastus medialis, VM; and vastus lateralis, VL). Eighteen university students (age: 24.1 ± 1.7 years, 9 females) underwent 7 weeks of parallel squat training (2 days/week) preceded by a 2-week familiarization period. Squat strength (1RM) and cross-sectional area (CSA) of four quadriceps muscles were assessed at baseline and at the end of the study. At posttest, 1RM and CSA of quadriceps muscles significantly increased (*p* < 0.01), with moderate-to-large effect (ES = 1.25–2.11) for 1RM (8.33 ± 6.64 kg), VM CSA (0.12 ± 0.08 cm^2^), and VL CSA (0.19 ± 0.09 cm^2^) and small effect (ES = 0.89–1.13) for RF CSA (0.17 ± 0.15 cm^2^) and VI CSA (0.16 ± 0.18 cm^2^). No significant differences were found in the changes of CSA between muscles (F = 0.638, *p* = 0.593). However, the squat 1RM gain was significantly associated only with the changes in CSA of the VL muscle (r = 0.717, *p* < 0.001). The parallel squat resulted in significant growth of all quadriceps muscles. However, the novelty of this study is that the increase in strength is associated only with hypertrophy of the VL muscle.

## 1. Introduction

The squat is one of the most popular and important exercises for the development of strength and power and is considered as an essential tool in strength and conditioning training programming [1]. Although the squat is a multi-joint exercise (involving the hip, knee, and ankle joints), it is generally believed that the quadriceps femoris (QF) muscles (rectus femoris, RF; vastus intermedius, VI; vastus medialis, VM; and vastus lateralis, VL) make the greatest contribution to the squat motion [2,3]. However, due to the bi-articular nature of RF (hip flexion and knee extension), it has been postulated that closed-chain exercises (squat, leg press, etc.) preferentially impact the vasti muscles among knee extensors [4,5,6,7]. This is also confirmed by electromyographic studies [5,8,9] in which RF displayed lower activity compared to VL and VM muscles. Moreover, a recent cross-sectional study by Kojic and coworkers [10] showed that parallel squat strength was significantly related only to the cross-sectional area (CSA) of the vasti muscles, while the contribution of RF to the squat strength appeared to be negligible. This leads to the logical conclusion that squat resistance training (RT) causes intermuscular hypertrophy of the QF.

Interestingly, there are only limited number of studies that have investigated how intermuscular hypertrophy of QF regions (i.e., RF, VL, VM, and VI) is affected by squat RT, and their findings are mostly inconclusive. For example, Kubo, Ikebukuro, and Yata [11] compared the effects of squat training with different depths on lower limb muscle volume. They observed an increase in the volume of the gluteal and three vasti muscles; however, the hypertrophy of RF was not significant for either squat regimen. Similar findings were noted by Earp et al. [12] and Zabaleta-Korta et al. [13], where parallel squat RT resulted only in an increase in CSA of the vasti muscles, suggesting that for targeting the RF muscle, single-joint exercises may be a better option [14]. However, some of the recent studies have reported opposite results [15,16,17]. Wilson et al. [16] investigated the effects of different RT exercise prescriptions (squat vs. deadlift vs. hip thrust) on VL and RF muscle thickness and found that all training modalities increased the size of tested muscles. Accordingly, Bjørnsen et al. [15] observed significant hypertrophy of all QF muscles following front squat RT. Fonseca et al. [18], surprisingly, found no significant growth in the VM and RF muscles although the increase in CSA from RF was close to statistical significance (*p* = 0.058). 

It should be noted that the majority of the aforementioned literature primarily investigated the effectiveness of different exercise protocols and RT variables, such as range of motion [11], contraction velocity [12,17], or exercise selection [13,15,18], rather than comparing hypertrophic responses between quadriceps muscles at the individual level. Therefore, the current study aimed to investigate (i) how RT with parallel back squat exercise affects the CSA of different regions of the quadriceps femoris and (ii) whether changes in the CSA of QF muscles can explain squat strength gains. We hypothesized that the CSA increase of the vasti (VM, VL, and VI) muscles would be significantly greater following the experimental protocol compared to RF and that hypertrophy of VL and VM would be significantly related to squat strength gains.

## 2. Materials and Methods

### 2.1. Experimental Design

This is a controlled study in which the primary aim was to investigate intermuscular differences among quadriceps muscles following squat training. The RT protocol included 7 weeks of parallel back squat training preceded by a 2-week familiarization period. Squat strength (one-repetition maximum—1RM) and CSA of four quadriceps muscles were assessed 2 days before the first training session and 2 days after the last training session. The flowchart of the experimental procedure is presented in Figure 1.

### 2.2. Subjects

The sample included 18 students from the University of Belgrade (9 males, 9 females, age: 24.00 ± 1.70 years, height: 1.75 ± 0.07 m, weight: 69.50 ± 10.48 kg, muscle mass: 32.12 ± 4.30 kg, body fat: 17.03 ± 3.86) who were healthy, had not participated in RT activities in the past 8 months, and had no chronic musculoskeletal diseases or injuries that could compromise the results of the present study. They were fully informed of the experimental procedures and potential risks and signed a written informed consent prior to participation in the study. Subjects were instructed to maintain their usual diet during the experimental period, to avoid ingestions of dietary supplements, and to refrain from any kind of physical activity for at least 48 h prior to testing. Participants were physically active as part of their normal academic curriculum, which included six to eight hours of low- to high-intensity exercise per week. They had no chronic diseases, cardiac problems, or recent musculoskeletal injuries. They were informed of the potential risks of the testing protocol used. Participants were blinded to the research question. The study was approved by the Institutional Ethics Committee (protocol code: 2316/19-2) and conducted in accordance with the Declaration of Helsinki and registered at ClinicalTrials.gov (accessed on 14 April 2021) (NCT04845295).

### 2.3. Testing

Body height was measured with a portable Martin’s anthropometer (Siber-Hegner, Zürich, Switzerland) with an accuracy of 0.1 cm. Body composition variables (skeletal muscle mass and body fat percentage) were measured with the In-Body 720 (Biospace Co., Seoul, Republic of Korea) using Direct Segmental Multifrequency—Bioelectrical Impedance Analysis (DSM–BIA method). 

A 1RM parallel squat test was performed to evaluate squat strength. The testing procedure was performed according to the standardized protocol [19]. Each participant had 5 attempts to lift the maximum weight. Inter-trial rests were set at 3 min. Participants were instructed to avoid any form of physical activity for at least 48 h prior to testing. 

The CSA of four quadriceps muscles (VM, VL, VI, and RF) was assessed using an ultrasound machine (Siemens Antares, Erlangen, Germany) with a variable high-frequency transducer (from 7 MHz to 13 MHz) and an image-fitting technique previously validated by our group [10,20]. Briefly, the RF CSA was measured at the level of three-fifths of the distance between the spina iliaca anterior superior and superior patellar border, whereas the CSA of VI was measured at the halfway point between the spina iliaca anterior superior and the proximal border of the patella. The CSA of the VL muscle was measured at 50% of the femoral length, defined as the 50% distance between the greater trochanter and the lateral condyle. The VM was measured at the level of the distal portion above the medial side of the patella. Image acquisition was performed by an experienced and previously trained radiologist. Subjects individually chose their dominant leg based on the question “Which leg would you use to shoot the ball?”, and this variable was used for the analysis [10]. The tests were performed by the same experienced researchers in pretest and posttest. 

### 2.4. Squat Training

Parallel barbell squat training (Appendix A) was performed in two sessions per week at the same time of day (1:00–3:00 p.m.) and with a rest of at least 48 h between sessions on the same days of the week. Each training session was preceded by a standardized warm-up routine performed by a licensed coach. The exercise was performed with a straight bar positioned above the acromion (i.e., high-bar position), with the feet shoulder-width apart and the toes pointed forward or slightly outward. The mentioned position was the same for all subjects in all training sessions. The range of motion of the exercise included a full concentric movement (up to the vertical position); during the eccentric phase, the movement was performed until the femurs were parallel to the floor, when the trochanter major and the lateral epicondyle of the femur were at the same level [10,20]. The height of the parallel squat was measured for each participant individually. An elastic band was then stretched, which participants had to touch during the squat to achieve the required angle of 90 degrees at the knee. All participants were able to achieve adequate squat depth using the same technique. During the first three weeks of the training intervention, participants performed exercise at ~60% of the 1RM in 3 sets; during the next four weeks, the load was set at 70% of the 1RM in 4 sets. Accordingly, the load volume (repetitions × sets × weight lifted) was recorded in the first and fourth weeks of the training intervention (i.e., 2017.6 kg and 2890.7 kg, respectively). All repetitions were performed until muscular failure. The rest between sets was 2 min. 

### 2.5. Statistics

The Shapiro–Wilk test was used to assess the normality of the distribution, whereas the homogeneity of the variances and the homogeneity of the regression slopes were tested by Levene’s test and by an interaction between the covariate and the independent variable, respectively. The reliability of the strength and ultrasound measurements was reported previously [20]. Baseline differences in CSA between measured muscles were tested by a one-way ANOVA. Because we found significant differences in CSA between QF muscles, a repeated-measures ANCOVA (using baseline values as covariates) was used to examine differences in the pre-to-post changes in 1RM and CSA for each quadriceps muscle and changes in CSA between muscles. If ANCOVA showed statistical significance, differences were further estimated using the Bonferroni post hoc test. Effect sizes (ES) were determined using G-power software (University of Kiel, Kiel, Germany, version 3.1) based on the recommendations proposed by Rhea [21] for untrained individuals; ES was considered trivial: <0.50, small: 0.50–1.25, moderate: 1.25–1.90, and large: >2.0.

In addition, pre- to post-training percentage changes (Δ) for the variables 1RM and CSA were calculated for each study participant. To examine whether changes in the CSA of the quadriceps muscles could explain the squat strength gains, linear regression was applied using Δ1RM as the dependent variable and ΔCSA of each quadriceps muscle as predictors. The strength of Pearson correlation (r) was classified according to the following model of Hopkins et al. [22]: small (0.1), moderate (0.3), large (0.5), very large (0.7), and extremely large (0.9). Statistical significance was set at *p* < 0.05. All statistical procedures were analyzed using SPSS version 20.0 (SPSS Inc., Chicago, IL, USA).

## 3. Results

1RM and CSA of quadriceps muscles increased significantly (all *p* < 0.01), with a moderate effect for 1RM (∆ 8.33 ± 6.64 kg, 95% CI: 5.03–11.64), VM CSA (∆ 0.12 ± 0.08 cm^2^, 95% CI: 0.08–0.16), and VL CSA (∆ 0.19 ± 0.09 cm^2^, 95% CI: 0.08–0.18) and a small effect for RF CSA (∆ 0.17 ± 0.15 cm^2^, 95% CI: 0.09–0.25) and VI CSA (∆ 0.16 ± 0.18 cm^2^, 95% CI: 0.08–0.16). At posttest, no significant differences in CSA changes were found between muscles (F = 0.638, *p* = 0.593, ES = 0.17) (Table 1, Figure 2).

Regression analysis identified ΔVL CSA as the only significant predictor of Δ1RM (r = 0.758, *p* < 0.001), with VL hypertrophy explaining approximately 55% (adj. R2 = 0.548, *p* < 0.001) of the variance in strength gain (Figure 3). On the other hand, the ΔCSA of RF (r = 0.006, *p* = 0.981), VI (r = 0.465, *p* = 0.052), and VM (r = 0.227, *p* = 0.365) were not significantly related to the Δ1RM.

## 4. Discussion

This study investigated the effectiveness of squat RT in promoting quadriceps hypertrophy at the individual level and also examined whether changes in the CSA of QF muscles were related to gains in squat strength. The main findings revealed that: (i) squat RT significantly increased the CSA of four knee extensors, with no differences between them, and (ii) the gain in squat strength was only related to the hypertrophy of VL muscle.

Contrary to our hypothesis, the present results demonstrate that the parallel squat is an effective training tool to induce a significant hypertrophic response of all QF muscles (i.e., the three vasti muscles and RF). Furthermore, ANCOVA analysis revealed no differences between muscles in terms of changes in CSA following the training intervention. This is an important finding considering previous assumptions that squat exercise does not sufficiently target RF among knee extensors [4,5,10]. However, it should be noted that the foundation of this concept has been preferably based on the lower electrical activity of RF compared to the vasti muscles during squats although a greater electromyographic response does not imply a greater hypertrophic potential [23]. 

Compared to previous longitudinal studies, our results corroborate well with some of the past reports [15,16,17] in which squatting RT resulted in equal hypertrophy among QF muscles. In contrast, no significant growth of RF was observed in other studies [11,12,13]. These conflicting findings may be related to several factors, including different methodological approaches (squat technique, diagnostic imaging, specific measurement region) or genetic predisposition. In the present study, the barbell was placed at the level of the acromion during the squat (i.e., in a high-bar position), whereas other studies used a low-bar version of the squat or did not mention the barbell position [11,12,15,16]. To increase the validity of the presented data and reduce the aforementioned confounding factors, all subjects were instructed to perform the squat using the same technique (same foot width, same foot angle, and same barbell position) as trained during the familiarization period. Recently, it was demonstrated that the position of the barbell highly affects thigh muscles activity, with high-bar squats eliciting greater involvement of the RF muscle compared to the low-bar position [24,25]. From a biomechanical perspective, this seems reasonable considering that high-bar squats are defined by a more upright torso, greater knee flexion, and less hip flexion compared to the low-bar variation, resulting in increased moment arm on the thigh muscles [26]. Therefore, a high-bar squat position could be responsible to some degree for the significant growth of the RF in the current study. 

Note that, although we found no differences between QF muscles in terms of hypertrophy, the magnitude of the effect favored VL and VM over VI and RF muscles (i.e., moderate vs. low ES). A closer inspection of the individual changes also reveals that some subjects greatly increased the CSA of VI and RF, whereas this increase was minor or even regressed in others (Figure 2). On the other hand, lower variability is observed in the hypertrophy of the VL and VM muscles. It can be concluded that squat-induced hypertrophy of the RF and VI muscles depends on intrinsic biological factors (related to individual characteristics of the subject), while VL and VM generally show uniform growth. In other words, squat exercise leads to uniform development of VL and VM, while some individuals may superiorly develop VI and RF. However, this still needs future research. 

The novelty of the present study is the fact that out of the four predictor muscles, only the ∆VL CSA correlated with changes in squat strength and explained about 50% of the ∆1RM variance. Thus, it appears that parallel squat RT leads to an increase in the size of all QF muscles, and still, only the hypertrophy of VL contributes to the squat strength gains. On one hand, this can be expected, as previous cross-sectional studies have identified VL as the important predictor of squat 1RM [10,27]. On the other hand, these results are quite surprising considering that our research group [10] has recently shown that the CSA of the VM muscle has the strongest association with strength in the parallel squat. However, a positive indicator of the above study is that VL CSA had a slightly higher correlation with external load in the parallel squat than in the deep squat. Nevertheless, a possible explanation may lie in the nature of the study itself and that results from cross-sectional studies may not translate into a longitudinal context, as is the case when examining relationships between muscle size and strength or RT-induced hypertrophy and strength gains [28]. In other words, although the size of the vasti muscles strongly determines the baseline squat performance, the further increase in squat strength is mainly the result of the morphological adaptations of VL. Furthermore, our results are consistent with the study by Wells et al. [29], in which lower-body RT resulted in an increase in VL muscle thickness that was significantly (r ≈ 0.6) associated with squat strength gains. Akagi et al. [17] also observed significant correlations (r ≈ 0.6) between squatting-induced increase in volume of the VL and VM muscles and peak torque of the knee extensors. The reason for the different results regarding the correlation between the ∆VM CSA and the increase in strength is probably due to the different methodology applied (isometric contraction vs. ∆1RM). From a hypertrophy perspective, changes in muscle strength are highly dependent on architectural adaptations (i.e., changes in pennation angle and fascicle length), where increase in pennation angle allows more contractile material to be packed within a given muscle, resulting in a greater capacity to produce force. Conversely, an increase in fascicle length as a result of the adding sarcomeres in series improves the maximum shortening velocities of the muscle [30]. In particular, VL has a large number of parallel force-producing contractile units, and lower-body RT usually provokes an increase in VL CSA mainly through an increase in the pennation angle rather than fascicle length [30,31,32]. On the other hand, a greater increase in fascicle length has been observed in RF and VM muscles [30,31]. In this context, it can be speculated that QF muscles do not show homogeneous architectural adaptations after squat RT, which in turn leads to a different correlation level with the changes in squat strength. Considering that the squat is a complex multi-joint exercise, the rest of the variance could be explained by neural adaptations (i.e., intra- and intermuscular coordination, motor unit firing frequency, etc.) and hypertrophy of the hip extensors [11,33]. 

Finally, we are aware that our study has some limitations that we can report. First, we measured the CSA of the QF at only one site. Considering that skeletal muscles may exhibit intramuscular (proximal, central, distal) hypertrophy in addition to intermuscular hypertrophy [30], we cannot rule out the possibility that greater growth occurred in other muscle regions. Second, although we advised study participants to maintain their usual dietary and exercise patterns, we did not systematically track their exercise or dietary behaviors, which could have hypothetically influenced the results obtained. In addition, it should be noted that the current results refer to the high-bar parallel back squat and should not be generalized to other squat variations (e.g., low-bar squat, front squat, etc.). Finally, it should be noted that correlational analyzes do not always imply direct causal evidence [34]; thus, our conclusions should be considered as an associative rather than a causal relationship between squat-induced hypertrophy and strength.

## 5. Conclusions

In conclusion, the parallel barbell back squat induces an equivalent hypertrophic effect on all quadriceps muscles when the squat technique is performed with a high bar placement with shoulder-width feet alignment. However, the novelty of this study is reflected in the fact that only hypertrophy of VL significantly contributes to the gains in the squat 1RM and explains about 50% of the variance in squat strength.

## Figures and Tables

**Figure 1 ijerph-19-16226-f001:**
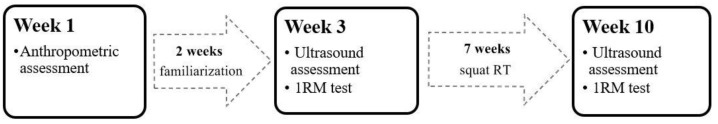
Flowchart of the experimental design.

**Figure 2 ijerph-19-16226-f002:**
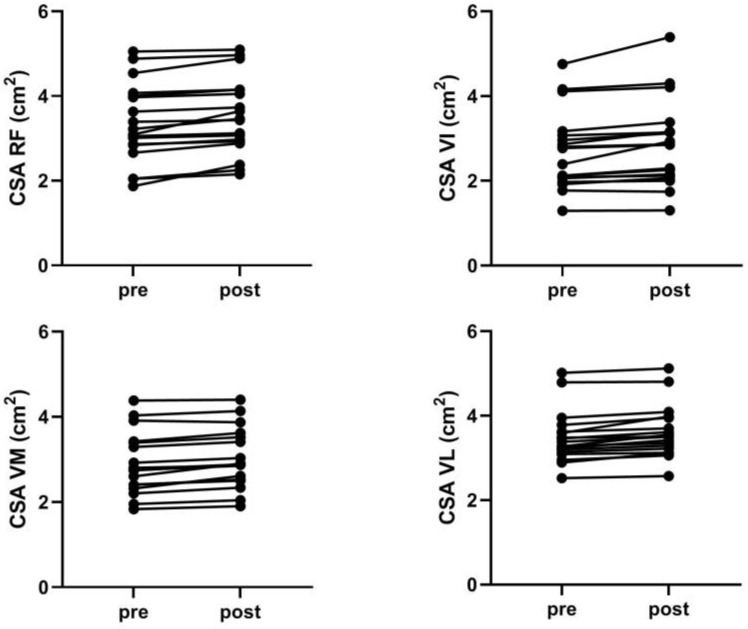
Individual pre-to-post changes in cross-sectional area (CSA) of rectus femoris (RF), vastus intermedius (VI), vastus medialis (VM), and vastus lateralis (VL).

**Figure 3 ijerph-19-16226-f003:**
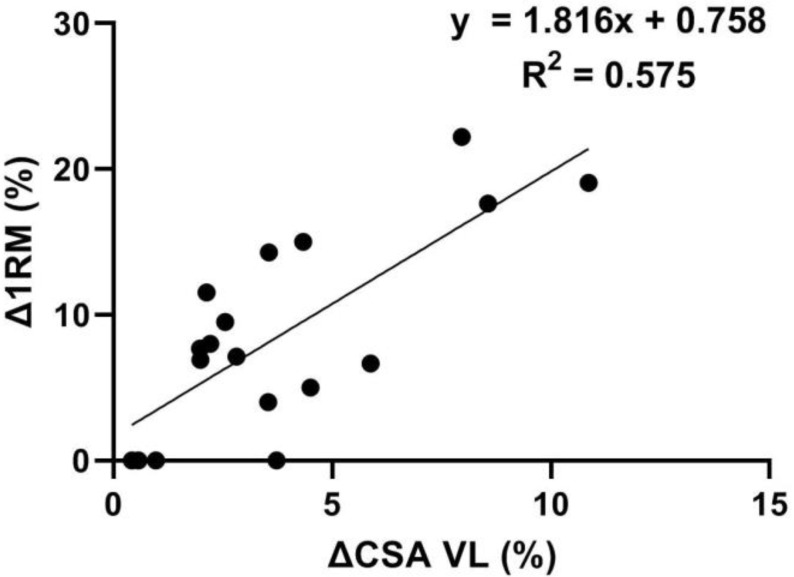
Relationship between relative changes (∆) in squat strength (1RM) and cross-sectional area (CSA) of vastus lateralis muscle.

**Table 1 ijerph-19-16226-t001:** Pre-to-post changes in 1RM and CSA of quadriceps muscles.

Variable	Pre	Post	*p*	ES
1RM (kg)	95.55 ± 24.00	103.89 ± 26.98	<0.001	1.25
RF CSA (cm^2^)	3.35 ± 0.93	3.50 ± 0.90	<0.001	1.13
VI CSA (cm^2^)	2.69 ± 0.92	2.84 ± 1.02	=0.001	0.89
VM CSA (cm^2^)	2.89 ± 0.71	3.01 ± 0.69	<0.001	1.50
VL CSA (cm^2^)	3.48 ± 0.62	3.62 ± 0.62	<0.001	2.11

1RM, one-repetition maximum; RF, rectus femoris; VI, vastus intermedius; VM, vastus medialis; VL, vastus lateralis; CSA, cross-sectional area.

## Data Availability

The data presented in this study are available on request from the corresponding author.

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
