# Peer review of "Does Back Squat Exercise Lead to Regional Hypertrophy among Quadriceps Femoris Muscles?"

_ijerph, 2022, doi:10.3390/ijerph192316226_

Round 1
Reviewer 1 Report
The authors presented interesting results with well-designed research study and well-presented results (hypertrophic responses at the individual quadriceps muscle level).
However, I am adding few minor comments:
- Please include that there are also strength gains on neuromuscular level (not only CSA)
- why did you choose direct testing of 1RM, and not recalculating 1RM from several repetitions?
- why did you choose 60% and 70% of 1RM for training load (hypertrophic response)?
Reviewer 2 Report
The current research examined if squat training induces inhomogeneous growth among quadriceps femoris muscles or not. I read the paper line by line and I congratulate the authors wrote a nice paper. Although the paper was well written, I have great doubts about whether the results of an increase in CSA of quadriceps muscles can only be related to squat training or not, because there is no control group in the current research design which needs to be backed up scientifically. Subjects were instructed to maintain their normal diet however did they obey this ? were the macro nutrient intakes of participants recorded during 7 weeks? How can you be sure about some participants increase their protein intake and others not ? did you monitor the physical activity or exercise regimens of participants during 7 weeks ? beacuse they are university student (probably sport science), is not it possible that they participate courses that increase their physical activity level which may induce hypertropic adaptations ? if this study had control group from same faculty, we can be sure that results related to CSA improvement of quadriceps is of squat training. Authors needs to clarify these points.
In the summary, line 17: data of males is missed
Line 44: authors concluded that squat resistance training (RT) causes inhomogeneous hypertrophy of the QF. Then why need this study ? there are also lots of study that investigate this topic (squat training and quadriceps muscles (VL, VM, VI, RF) hyprtrophy.
Line 53: …….. single joint exercises may be a better option, okey but you did not investigate single joint exercise’s effect, what is the purpose of this phrase ? you line the studies in this topi (11,12,13) however, what is your differences from them ? they compare the full squat training groups either control group or other training regimens but you only have training group, it was expected that you have more from that studies but in reverse you have less and nothing new ?? please enlight me if I miss something.
Line 54-59: these studies’s (15-18) results cannot be concluded as confusing and the influence of squat RT on intermuscular hypertro- 60 phy of the quadriceps femoris remains an open question. Differences in these studies can stem from training and nutritional status of participants or gender and this moderating factors was not handle by your studies so I think if you dont fill this gap, is there a need sentence of line 60-61?
Line 62-66: although the studies (11-18) investigated effects of different RT designs on hypertrophy of QF muscles, they also analyzed not only between-subject design but also wihtin-subject design. They also analyzed group means so investigate each individual result.
In the intro, I am not much satifsy why need this study that have less variables than the studies in this topic ?
Method section was reported sufficiently.
Line 134 : why you choose repetition to failure protocol ?
Line 165: table 1. 3th line (VI CSA), lack of “<”0.001
Line 216: is this be related to gender differences of subject (9 females, 9 males) ??
Ä°t could be better to present by gender and seperate for example the figüre 2 as figure 2a (males), figure 2b (females).
I think there must be a paragrapgh related to gender differences and similarities in hypertropic adaptations.
Line 265-266: Yes, you report lack of tracking diet as a limitations but it is enough to hinder this factor that may affect hypertrophic adaptations ?
Reviewer 3 Report
I am grateful for the opportunity to review this manuscript titled "Does parallel squat resistance training lead to inhomogeneous growth among quadriceps femoris muscles?”. This study aimed to test if squat exercise induces inhomogeneous growth among quadriceps femoris muscles (rectus femoris, vastus intermedius, vastus medialis and vastus lateralis) in university students without recent experience in resistance training. The data collected in this study may affirm or expand on available literature.
This study is of interest to the IJERPH readers and seems to provide some new findings, applicable to the fields of training. However, the points mentioned in the “Specific comments” section below should be considered and the manuscript amended accordingly before being considered for publication.
Specific comments
Title and abstract
1. The keyword 'resistance training' should not appear in the title, besides that it sounds redundant, with the squat exercise being intuited that it is resistance training, besides it is indicated in the abstract. Typing 'back squat exercise' should suffice.
2. As indicated in the previous point, it should be specified what type of squat is performed, since there may be many variants to be performed, for which reason it has not been written in a specific way (e.g., back squat).
3. The term 'inhomogeneous growth' does not seem intuitive, and there are not many studies that term it in that way. It might be more intuitive/standardized to write 'regional hypertrophy' or 'specific hypertrophy'.
4. The aim should be better specified, it is the same as the title.
5. When describing the participants, the authors only mentioned the number of females and not the number of men (‘9 females’).
Introduction
6. The introduction is well written, and the authors have highlighted in the introduction the contribution of their work to the area, but the authors should be clearer with the terms, they vary a lot between ‘growth’, ‘hypertrophy’, ‘inhomogeneous’, and ‘specific’. They should use more concrete language.
Materials and Methods
2.1. Experimental design
7. Were the same researchers who performed the assessments pre and post? In addition, it is not mentioned if the experimenters were blinded to the research question.
2.2. Subjects
8. The age data does not match concerning the abstract (24.1 ± 1.7 / 24.00 ± 1.3).
2.3. Testing
9. It does not indicate which extremity is being measured, whether left/right, dominant/non-dominant...
2.4. Squat Training
10. It is not indicated if it is a barbell squat or with a Smith machine (on the other hand, it is specified in the 'Conclusions' section).
11. If it is a barbell squat, is it the best exercise to experiment with subjects unfamiliar with the back squat or strength training?
For the above reason, the technique and the lack of experience is a limiting factor when compiling the results since it is a complex exercise for the subjects to execute in the same way and reach muscular failure.
12. Why has not the fatigue of each subject been evaluated in each series performed? (For example, rate perceived exertion, RPE). For the same reason, the fact that the rest time is fixed (2 minutes) does not reflect a real recovery adapted to the effort of each participant, especially performing each series to muscular failure.
13. It could be interesting to show the performance of the squat exercise in images, to give the reader an illustrative idea of how the movement has been executed.
Discussion
14. More limitations should be included: the data is only valid for one type of squat; subjects have at least 8 months of detraining in resistance training
Round 2
Reviewer 2 Report
I thanks to the authors that they spend time on paper once again and revised the paper according to the reviewer suggestion, paper now is looking better.
Reviewer 3 Report
The authors have made the suggested changes, and the manuscript has improved considerably.